# Regional Differences in Gene Expression of Proliferating Human Choroidal Endothelial Cells

**Andrew C. Browning** [1], **Eugene P. Halligan** [2], **Elizabeth A. Stewart** [1], **Daniel C. Swan** [3], **Simon J. Cockell** [3] and **Winfried M. Amoaku** [1,*]

1. Academic Ophthalmology, Queen's Medical Centre, Division of Clinical Neurosciences, University of Nottingham, Nottingham NG7 2UH, UK; Andrew.browning@nhs.net (A.C.B.); Elizabeth.Stewart@exonate.com (E.A.S.)
2. Department of Infection and Immunology, 5th Floor, North Wing, St Thomas's Hospital, London SE1 7EH, UK; Eugene.Halligan@seracare.com
3. Bioinformatics Support Unit, Institute of Cell and Molecular Biosciences, Newcastle University, Newcastle upon Tyne NE2 4HH, UK; D.Swan@ncimb.com (D.C.S.); simon.cockell@ncl.ac.uk (S.J.C.)
* Correspondence: wma@nottingham.ac.uk or winfried.amoaku@nottingham.ac.uk; Tel.: +44-115-9249924 (ext. 64744)

**Abstract:** Choroidal diseases including inflammation and neovascularization seem to have predilection for different vascular beds. In order to improve our understanding of human macular choroidal angiogenic diseases, we investigate the differences in gene expression between matched human macular and peripheral inner choroidal endothelial cells (CEC) and matched human macular inner and outer CEC. The gene expression profiles of matched, unpassaged human macular and peripheral inner CEC and matched human unpassaged macular inner and outer CEC were conducted using Affymetrix GeneChip arrays. Selected differences in gene expression were validated by real-time-PCR and immunohistochemistry. No differences in probeset expression were demonstrated between inner CECs compared with peripheral inner CECs. In comparison, there was a difference of 1.6% of probesets when matched, unpassaged proliferating human macular inner CEC and macular outer CEC from the same donors were compared. Macular inner CECs demonstrated up-regulation of probesets involved in nervous system development, growth factors, *PLVAP*, and *collagen XVI*, while macular outer CECs demonstrated up-regulation of probesets involved in immune function and intracellular signalling. There was a marked homogeneity of human macular and peripheral inner CECs. This suggests that gene expression differences in inner CECs are not responsible for the site specific selectivity of choroidal neovascularisation. Variability was noted, however, in the gene expression of matched macular inner and outer CECs. This could be explained by the differences in the roles and microenvironments of the inner and outer choroid.

**Keywords:** choroidal endothelial cells; inner choroid; outer choroid; peripheral choroid; choroidal neovascularization; gene expression

## 1. Introduction

The human macula appears predisposed to a wide range of different sight threatening disorders, particularly, choroidal neovascularisation (CNV). The reason(s) for this predisposition remain largely unknown; however, it has been well recognised for many years that the ultrastructure of the macula area is different from the more peripheral areas of the posterior segment. More recently, investigators have examined the gene expression within different structures of the macula and compared it with their more peripheral counterparts in an attempt to explain this paradox [1–4].

Within the retina, there is a variable distribution of rods and cones across the posterior segment. Although the macula only occupies 1.4% of the retinal area, it contains 8.4% of all retinal cone figures and 3.4% of rods and 60% of ganglion cells [5]. Not surprisingly,

comparison of gene expression between macular and peripheral retina found a number of genes preferentially expressed in the macula that were highly expressed in cones and axons, while the retinal periphery appeared enriched for rod specific genes [1,6,7]. Whilst these differences in the gene expression within the neurosensory retina are associated with well documented and understood physiological functions, differences in the topographic distribution in sub-retinal cells and structures are less well understood. Ultrastructurally, retinal pigment epithelial (RPE) cells from the macula and the periphery differ in appearance, with those from the macula being more columnar [8] and containing more melanin than those in the periphery [2,9]. Several studies have investigated the variation in RPE gene expression across the posterior segment. [2,4]. Using laser microdissection to harvest areas of macular RPE and choroid in human donor eyes, Ishibashi [2] and Van Soest [4] independently identified a small number of genes that were differentially upregulated the macula. These authors suggested that this may represent a role for the RPE in Bruch's membrane turnover, and may contribute to the topographical differences found by other investigators in the composition and physical properties of Bruch's membrane and which in turn may lead to the increased disease susceptibility of the macula [10–12].

The ultrastructure of the choroid is well described, being composed of three distinct layers: an outer layer of large vessels (Haller's layer), a middle layer of medium sized vessels (Sattler's layer) and an internal layer adjacent to Bruch's Membrane composed of capillary vessels (choriocapillaris). Less is known regarding the heterogeneity of choroidal endothelial cells (CECs) and the role that this may play in the predisposition and pathogenesis of various choroidal disorders. As an example, the CNV seen in neovascular age-related macular degeneration (nAMD) usually begins within the inner choroid and almost always occurs within the macular area [13,14]. It is not known if the CECs in this sub-macular area are different to those in the periphery, thereby making them more susceptible to certain diseases, and whether they may respond differently to selective treatments. Alternatively, it may be that that the CECs are all the same and they are simply responding to secondary events brought about by the topographical differences of other cell types (retinal and RPE), structure (Bruch's membrane) or insults (UV light damage or trauma) that in turn brings about the site specificity of the disease.

The aim of this study was to compare the gene expression profiles of proliferating matched human sub-macular inner and outer CECs, and matched human sub-macular and peripheral inner CECs to determine whether gene expression profiles are localised to cells within different areas of the human choroid. Differences in gene expression would elucidate differences in the roles of ECs and microenvironments of the inner and outer choroid. These findings may help explain the topographical selectivity and mechanisms of choroidal neovascularization, and perhaps lead to the development of site specific treatments for choroidal vascular diseases including nAMD.

## 2. Materials and Methods

### 2.1. Isolation of CEC

The technique for the isolation of human macular inner CECs has been described previously [15]. For this investigation, matched 6 mm diameter samples of peripheral inner choroid (peripheral area nasal to the optic disc) and macular outer choroidal tissue were dissected, collected and treated in the same manner as described previously for the macular inner choroidal samples [15]. Briefly, 6 mm samples were removed from the appropriate areas of the posterior segments and transferred to Petri dishes where the retina was discarded and the retinal pigment epithelium (RPE) was removed by gentle brushing with a sterile spatula and by irrigation with sterile phosphate buffered saline (PBS). The choroid was the teased from the attached sclera and turned upside down. With the use of a dissecting microscope, the large outer choroidal vessels along with the adherent fibrous tissue were peeled off and placed in PBS. The remaining, relatively non-pigmented inner choroidal tissue was also placed in PBS. The samples were then cut into 1 mm pieces and washed 3 times in minimum essential medium (MEM) containing 30 mM

HEPES, 0.25 μg/mL amphotericin B, 100 μg/mL streptomycin, 50 μg/mL kanamycin and 30 μg/mL penicillin (isolation medium. The pieces were then incubated in 0.1% collagenase I (Sigma-Aldrich, St. Louis, MO, USA) in MEM for 2 h at 37 °C with frequent agitation. The collagenase was neutralized with MEM containing 10% foetal calf serum (Invitrogen, Paisley, UK) and the mixture was filtered through sterile 40 μm and then 20 μm filters (Millipore Ltd., Watford, UK). The eluate was centrifuged (75 g) and washed 3 times in isolated medium and resuspended in 1 mL of PBS/0.1%BSA. The endothelial cells were then isolated using anti-CD31 coated Dynabeads using the manufacturer's instructions (Dynal Ltd., Wirral, UK). The Dynabeads were resuspended in endothelial growth medium (EGM2-MV with hydrocortisone omitted) (Cambrex Biosciences, Wokingham, Berks, UK) and seeded onto fibronectin coated 35 mm culture dishes (Beckton Dickinson, Oxford, UK). The unpassaged cells were grown to 80% confluence before a small sample was removed for confirmation of purity or RNA extraction using a Qiagen RNeasy minikit (Qiagen, Crawley, UK) by following the manufacturer's instructions. The RNA was stored at −80 °C in microcentrifuge tubes until required for subsequent analysis. The research had the approval of the local research ethics committee (Nottingham Q1060301) and complied with the tenets of Helsinki Declaration for medical research involving humans.

### 2.2. Confirmation of EC Purity

The identity and purity of cells used in the microarray assays was confirmed prior to RNA extraction by demonstrating that greater than 99.5% of cells stained positive for factor VIII and CD31, and negatively stained with anti-rat alpha smooth muscle actin (aSMA) or fibroblast surface protein as previously described [15]. Briefly, samples of choroidal endothelial cells to be used for microarray analysis were taken and grown on 1% gelatin coated glass cover slips and then fixed in ice cold methanol at −20 °C for 20 min. A standard two stage immunofluorescence technique was applied by incubating the cells with primary antibodies against CD31 and vWF (Dako, Cambridgeshire, UK) for 4 h followed by secondary antibody after washing with PBS. The secondary antibody was rabbit anti-mouse F(ab')$_2$ fragment, FITC conjugated (1:20 dilution) for the CD31 stain and swine anti-rabbit F(ab')$_2$ fragment, FITC conjugate (1:20 dilution) for the vWF stain (Dako, Cambridgeshire, UK). The slides were mounted in glycerol containing DABCO (Sigma-Aldrich, St. Louis, MO, USA) and observed by confocal fluorescence microscopy (Leica TCS40D, Leica, Milton Keynes, UK).

### 2.3. RNA Extraction

The isolated endothelial cells from the 3 different intra-choroidal locations were propagated under identical conditions. The total RNA was extracted from primary cultures when they had reached approximately 80% confluence, using the Qiagen RNeasy minikit as directed (Qiagen, Crawley, UK). RNA integrity and quality was assessed using an Agilent 2100 Bioanalyser and RNA 6000 Nano kit (Agilent Technologies, Santa Clara, CA, USA). Total RNA was extracted from primary cultures of unpassaged unmatched endothelial cells when they had reached approximately 80% confluence, using the Qiagen RNeasy minikit as directed (Qiagen, Crawley, UK). Approximately 5 μg of total RNA was obtained from each 35 mm culture plate.

### 2.4. Microarray Analysis

Microarray analysis was performed using Affymetrix GeneChip® Human Genome U133 Plus 2.0 arrays (Affymetrix, High Wycombe, Bucks, UK) as previously described [16]. Briefly, the previously stored RNA was thawed, the concentration rechecked using the Nanodrop ND-1000 spectrophotometer. The RNA integrity and quality was assessed using an Agilent 2100 Bioanalyser and RNA 6000 nanokit (Agilent Technologies, Santa Clara, CA, USA) by following the manufacturer's instructions. A sample needed an RNA integrity index (RIN) greater than 8 to be taken forward for analysis. Biotinylated complementary RNA probes were prepared from the total RNA samples (1 μg) using the microarray target amplification kit

and the microarray target RNA synthesis kit (both Roche Applied Sciences, Burgess Hill, UK) following the manufacturer's instructions. Samples of fragmented cRNA were hybridized onto Affymetrix GeneChip human genome U133plus 2.0 arrays (Affymetrix, High Wycombe, Bucks, UK) along with control oligo B2 and eukaryotic hybridisation controls (*bioB*, *bioC*, *bioD* and *cre*) at 45 °C for 16 h. This was followed by the washing, staining and scanning of the arrays using a Fluidics Station 450, Affymetrix scanner 3000 and GCOS software (Affymetrix GeneChip protocols). The GCOS software (Affymetrix, Santa Clara, CA, USA) was used to monitor scanning and to convert the raw image files into intensity files ('.CEL').

### 2.5. Data Analysis

Affymetrix CEL files were imported into GeneSpring GX 11.0.1 and processed with the MAS5 algorithm for probe level quality control. Data was then normalized with GC-RMA to provide expression values. To identify differentially expressed genes between cell groups, a one way ANOVA was performed with Tukey-HSD post hoc testing, where the 3 unique regions were considered to be levels of the same factor. Benjamini–Hochberg false discovery rate control was applied to all probesets. Due to the small sample size (3 samples at each intra ocular location), the data was also subject to rank products analysis [17]. A difference in expression between probesets (macular inner choroidal ECs vs. macular outer choroidal ECs) with a corrected *p*-value of <0.05 and a fold change of greater than 2 in all samples from a particular location were considered to be statistically significant. The genes demonstrating significant differences between the datasets (macular inner vs. macular outer CECs) were exported directly to Ingenuity Pathway Analysis for canonical pathways was done using the following parameters: human genes, metabolic pathways, cell signaling, nervous system development, cellular growth and proliferation, immune response and cell morphology.

### 2.6. QPCR

Expression data from the microarray experiments was validated by TaqMan real-time PCR using the same samples as those used in the microarray experiments [16]. Genes with transcripts that demonstrated at least a 2-fold differential expression between ECs from the various sources on microarray were selected. Furthermore, selected probesets were thought to be relevant to a range of EC functions, including cell signaling, cell morphology, nervous system development, cell growth and proliferation, and miscellaneous functions. Briefly, cDNA from each of the samples was generated from 50 ng of total RNA using the Superscript III first strand synthesis system (Invitrogen). The resulting cDNA was subjected to real-time PCR reactions in triplicate using the manufacturer's TaqMan Universal mastermix kit protocol and the ABI PRISM 7000 sequence detection system. The expression of hypoxanthine–guanine phosphoribosyltransferase (HPRT) was chosen for normalisation. Analysis of the relative gene expression data was performed using the $\Delta\Delta$ Ct method. The crossing point or threshold (Ct) of each target gene was normalized against the HPRT housekeeping gene before the fold change was calculated, relative to the target gene expression in a different anatomical location.

### 2.7. Immunohistochemistry

Immunohistochemical validation of the results was undertaken by staining sections of choroid with relevant antibodies to a selection of proteins found to be upregulated at the gene expression level by microarray analysis. A 5 mm biopsy punch was used to remove cores of macula and peripheral choroid, retina was removed and the choroid/sclera core embedded in optimal cutting temperature (OCT) freezing compound (Leica, Germany) and frozen using liquid nitrogen. Using a cryostat (Leica), 7 µm sections were prepared and fixed with 100% acetone. Sections were blocked with 3% (*w/v*) bovine serum albumin (BSA) in phosphate buffered saline (PBS) and stained with primary antibodies targeted to brain derived neurotrophic factor [BDNF] (Abcam, Cambridge, UK; ab108383) or plasmalemma vesicle associated protein [PLVAP] (Abcam, ab81719), overnight, at 4 °C. Primary antibodies were

visualised using the appropriate secondary antibody conjugated to tetramethyl rhodamine iso-thiocyanate (TRITC) at 1:400 for 1 h at room temperature. Slides were counterstained with 4′,6-Diamidino-2-phenylindole (DAPI; 5 µg/mL; Santa Cruz Biotechnology Inc, Heidelberg, Germany) and examined on a fluorescence microscope (Olympus BX51) and imaged using Cell^F software (Olympus, Southend-on-Sea, UK). Each experiment was performed on multiple sections from at least 2 donors and comparative fluorescence was detected using the same settings. For haematoxylin and eosin (HE) staining serial 7 µm sections were fixed with 3% (*v*/*v*) acetic alcohol, washed in TBS then stained in haematoxylin for 1–2 min, washed in tap water for 1 min, Scott's Tap water for 1 min, then distilled water for 1 min. Sections were then stained with eosin Y for 1 min, then washed in tap water for 1 min, Sections were visualised used a Nikon TS100 light microscope and digital camera.

## 3. Results

Nine matched, un-passaged EC samples, representing 3 different intra-choroidal locations from three different donors were propagated under identical conditions. The 3 choroidal areas represented by the matched samples were: macular inner CEC, macular outer CEC and peripheral inner CEC (peripheral area nasal to the optic disc). The age, sex and time from death to the cells being placed in culture medium were as follows: 58, male, 28 h: 42, male, 43 h: 62, female, 36 h. All eyes were free of ocular disease, in particular, the macula, on examination with the dissecting microscope.

### 3.1. Confirmation of Human Endothelial Cell Identity

Samples of cells from all locations displayed a homogeneous cobblestone morphology with no evidence of cell contamination. Greater than 99.5% of the EC from each site demonstrated staining for factor VIII and CD31 prior to their use in the aforementioned experiments, confirming their identity as EC and their purity (Figure 1).

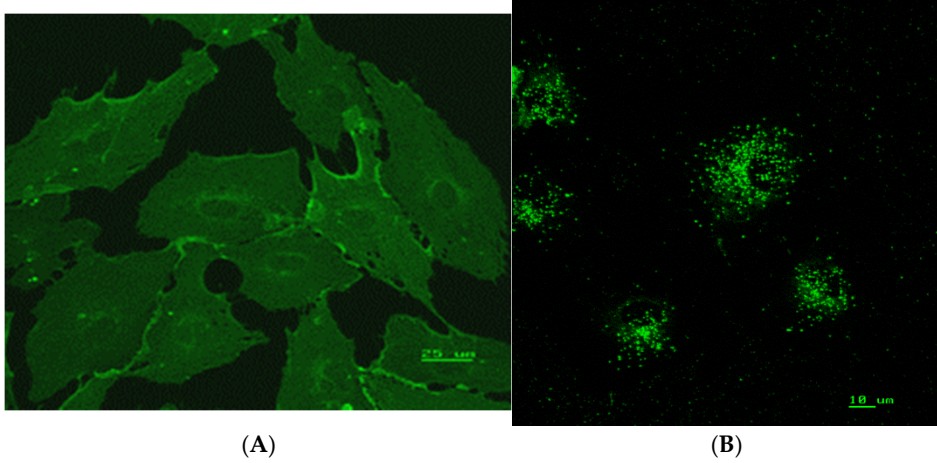

(**A**)             (**B**)

**Figure 1.** (**A**) Immunofluorescent photomicrograph of choroidal endothelial cells stained for CD 31. Predominance of staining occurs at the edge of the cells where density of CD31 is known to be highest. (**B**) Immunofluorescent photo micrograph of human choroidal endothelial cells showing dense staining for vWF.

### 3.2. Overview of Gene Expression Patterns

A total of 1.5–3.4 µg RNA was isolated from each 35 mm plate. All samples analysed had a RIN of between 9.5 and 9.Biotinylated cRNA probes were hybridised to Affymetrix GeneChip® Human Genome U133 Plus 2.0 arrays containing 47,000 transcripts, of which 38,500 were well characterised human genes. A total of 23,636 probesets passed quality control testing during analysis. Complete data is available at http://www.ncbi.nlm.nih.gov/geo/query/acc.cgi?token=gnavimkwbrarxqf&acc=GSE51246 (accessed on 27 September 2013).

Supplementary Figure S1 shows an RNA degradation plot for the nine individual arrays. While RNA degradation was evident, it was similar for all nine arrays. Supplementary Figure S2 shows the nine arrays plotted along the first and second principle components. The plot shows that the macula outer CECs from all three donors are grouped together, while the macular and peripheral inner CECs from each individual donor are positioned close together.

### 3.3. Proliferating Human Macular Inner CEC Versus Peripheral Inner CEC

Comparison of matched, un-passaged proliferating human macular inner CECs with peripheral inner CECs did not demonstrate any difference in gene expression. This study, therefore, demonstrated a striking homogeneity of gene expression between matched inner CECs derived from the macula and periphery.

### 3.4. Proliferating Human Macular Inner CEC Versus Macular Outer CEC

Comparison of matched, un-passaged proliferating human macular inner CEC with macular outer CEC revealed 386 probesets that were differentially expressed between the two cell types (1.6%). Of the 386 probesets, 151 were up-regulated in macular inner CEC, and 235 were up-regulated in macular outer CEC. Probesets for a diverse range of functions including nervous system development (BDNF, Neurofilament light and medium polypeptides), cell signalling (MAPK 11, Apelin receptor, Parvin beta, diacylglycerol kinase and MAPK binding protein 1 and WNT5A), cellular growth and proliferation (VEGF, HGF, CXCL12, TGF Beta1 and MMP10), immune response (MHC class II alpha and HLA DR B1 and CD200) and cell morphology (Keratin 19, Collagen 11 alpha subunit and Collagen 15 alpha subunit and PLVAP) were found to be differentially expressed. Lists of the most highly expressed probesets reaching statistical significance are shown in Tables 1 and 2, and Figure 2 shows a heat map of the differential probeset expression reaching statistical significance with reference to human macular inner and outer CEC.

**Table 1.** The top 100 probesets reaching differential expression of 2.0 or greater and corrected *p*-value less than 0.05 in human macular inner CEC compared with macular outer CEC.

| Gene Title | Fold Change | Uncorrected *p*-Value | Corrected *p*-Value |
|---|---|---|---|
| insulin-like growth factor binding protein 3 | 11.1 | 0 | 0 |
| neurofilament, medium polypeptide | 10.4 | 0 | 0 |
| brain-derived neurotrophic factor | 8.8 | 0 | 0 |
| platelet-derived growth factor receptor, alpha polypeptide | 8.8 | 0 | 0.0029 |
| keratin 19 | 7.3 | 0 | 0.0033 |
| pleckstrin homology-like domain, family A, member 2 | 7.1 | 0 | 0 |
| solute carrier family 6 (neutral amino acid transporter) | 6.6 | 0 | 0 |
| popeye domain containing 3 | 6.0 | 0 | 0.0031 |
| olfactomedin-like 3 | 5.9 | 0 | 0.0038 |
| SIX homeobox 2 | 5.6 | 0 | 0.0035 |
| neurofilament, light polypeptide | 5.5 | 0 | 0.0033 |
| adipocyte-specific adhesion molecule | 5.2 | 0 | 0.0027 |
| CCAAT/enhancer binding protein (C/EBP), beta | 5.2 | 0 | 0.0107 |
| mesoderm specific transcript homolog (mouse) | 5.1 | 0 | 0.0036 |
| ADAM metallopeptidase with thrombospondin type 1 | 5.1 | 0 | 0.005 |
| mannosyl (alpha-1,3-)-glycoprotein | 5.1 | 0 | 0.0029 |
| keratin associated protein 2–4 | 5.0 | 0 | 0.0044 |
| chromosome 5 open reading frame 23 | 4.9 | 0 | 0.004 |
| vascular endothelial growth factor A | 4.9 | 0 | 0.0197 |
| brain expressed, X-linked 1 | 4.8 | 0 | 0.0113 |
| GLI pathogenesis-related 1 | 4.7 | 0 | 0.0096 |
| lysophosphatidic acid receptor 1 | 4.7 | 0 | 0.0177 |
| Aldehyde dehydrogenase 1 family, member L2 | 4.6 | 0 | 0.0264 |

**Table 1.** *Conts.*

| Gene Title | Fold Change | Uncorrected *p*-Value | Corrected *p*-Value |
| --- | --- | --- | --- |
| collagen, type XI, alpha 1 | 4.4 | $1.0 \times 10^{-4}$ | 0.0388 |
| pleckstrin and Sec7 domain containing 3 | 4.4 | 0 | 0.0037 |
| carbonic anhydrase XII | 4.4 | $1.0 \times 10^{-4}$ | 0.0393 |
| procollagen C-endopeptidase enhancer 2 | 4.3 | 0 | 0.0272 |
| ABI family, member 3 (NESH) binding protein | 4.3 | 0 | 0.0269 |
| phosphodiesterase 5A, cGMP-specific | 4.2 | 0 | 0.0107 |
| PNMA-like 1 | 4.1 | 0 | 0.0205 |
| tissue factor pathway inhibitor 2 | 4.1 | $1.0 \times 10^{-4}$ | 0.0408 |
| glycine receptor, beta | 4.0 | 0 | 0.034 |
| fin bud initiation factor homolog (zebrafish) | 4.0 | 0 | 0.013 |
| wingless-type MMTV integration site family, member 5A | 4.0 | 0 | 0.0038 |
| sphingosine-1-phosphate receptor 3 | 4.0 | 0 | 0.0298 |
| carbonic anhydrase XII | 3.9 | 0 | 0.0171 |
| forkhead box F2 | 3.9 | 0 | 0.0321 |
| MSTP150 | 3.9 | $2..0 \times 10^{-4}$ | 0.0495 |
| Ras-related C3 botulinum toxin substrate 1 | 3.9 | $1.0 \times 10^{-4}$ | 0.0406 |
| frequenin homolog (Drosophila) | 3.8 | 0 | 0.0207 |
| glycine receptor, beta | 3.8 | $1.0 \times 10^{-4}$ | 0.0405 |
| vestigial like 3 (Drosophila) | 3.8 | 0 | 0.0259 |
| TP53 regulating kinase | 3.8 | $1.0 \times 10^{-4}$ | 0.0421 |
| chromosome 6 open reading frame 141 | 3.8 | 0 | 0.0095 |
| calmegin | 3.7 | $1.0 \times 10^{-4}$ | 0.0359 |
| carbonic anhydrase XII | 3.7 | 0 | 0.0152 |
| hypothetical protein LOC100130506 | 3.7 | 0 | 0.0112 |
| ectonucleotide pyrophosphatase/phosphodiesterase 2 | 3.7 | $1.0 \times 10^{-4}$ | 0.0437 |
| protocadherin 18 | 3.6 | 0 | 0.0295 |
| leucine rich repeat (in FLII) interacting protein 1 | 3.6 | 0 | 0.0155 |
| hepatocyte growth factor (hepapoietin A; scatter factor) | 3.6 | 0 | 0.0091 |
| protein phosphatase 1, regulatory (inhibitor) subunit 14B | 3.6 | $1.0 \times 10^{-4}$ | 0.0486 |
| golgi autoantigen, golgin subfamily a, 8A | 3.6 | 0 | 0.0159 |
| lymphoid-restricted membrane protein | 3.5 | 0 | 0.0341 |
| sulfatase 1 | 3.4 | $1.0 \times 10^{-4}$ | 0.0471 |
| dermatan sulfate epimerase-like | 3.4 | 0 | 0.0116 |
| secretogranin II (chromogranin C) | 3.4 | $1.0 \times 10^{-4}$ | 0.0408 |
| poliovirus receptor-related 3 | 3.4 | 0 | 0.026 |
| metastasis associated lung adenocarcinoma transcript 1 | 3.4 | 0 | 0.0205 |
| folate receptor 1 (adult) | 3.4 | 0 | 0.021 |
| transforming growth factor, beta receptor 1 | 3.4 | $1.0 \times 10^{-4}$ | 0.0386 |
| spermatogenesis associated 18 homolog (rat) | 3.4 | $1.0 \times 10^{-4}$ | 0.0429 |
| tissue factor pathway inhibitor 2 | 3.4 | 0 | 0.0337 |
| emopamil binding protein-like | 3.4 | $1.0 \times 10^{-4}$ | 0.0354 |
| aldehyde dehydrogenase 1 family, member A3 | 3.4 | 0 | 0.0283 |
| centromere protein V | 3.4 | 0 | 0.0256 |
| interferon-induced protein with tetratricopeptide repeats 1 | 3.4 | 0 | 0.0332 |
| lactamase, beta | 3.4 | 0 | 0.0183 |
| 5′-nucleotidase, ecto (CD73) | 3.3 | $1.0 \times 10^{-4}$ | 0.0426 |
| tumor necrosis factor (ligand) superfamily, member 15 | 3.3 | 0 | 0.0203 |
| mitogen-activated protein kinase kinase kinase kinase 4 | 3.3 | $1.0 \times 10^{-4}$ | 0.0448 |
| chromosome 9 open reading frame 40 | 3.3 | 0 | 0.0319 |
| neuropilin (NRP) and tolloid (TLL)-like 2 | 3.3 | $1.0 \times 10^{-4}$ | 0.0347 |
| family with sequence similarity 13, member B | 3.3 | $1.0 \times 10^{-4}$ | 0.041 |
| aspartyl-tRNA synthetase | 3.2 | 0 | 0.0195 |
| Chromodomain helicase DNA binding protein 2 | 3.2 | $1.0 \times 10^{-4}$ | 0.0404 |
| paternally expressed 10 | 3.2 | 0 | 0.0311 |
| versican | 3.1 | 0 | 0.0298 |
| metastasis associated lung adenocarcinoma transcript 1 | 3.1 | 0 | 0.0337 |
| nephronophthisis 3 (adolescent) | 3.1 | 0 | 0.0265 |
| carboxymethylenebutenolidase homolog (Pseudomonas) | 3.1 | 0 | 0.0293 |
| hypothetical LOC100128822 | 3.1 | $1.0 \times 10^{-4}$ | 0.0434 |
| phosphoglycolate phosphatase | 3.1 | 0 | 0.0262 |
| prostaglandin-endoperoxide synthase 2 | 3.0 | 0 | 0.0227 |
| retinol dehydrogenase 14 (all-trans/9-cis/11-cis) | 3.0 | 0 | 0.0267 |
| carbohydrate sulfotransferase 7 | 3.0 | $1.0 \times 10^{-4}$ | 0.0444 |
| proline rich 16 | 3.0 | $2.0 \times 10^{-4}$ | 0.0498 |

**Table 1.** *Conts.*

| Gene Title | Fold Change | Uncorrected *p*-Value | Corrected *p*-Value |
|---|---|---|---|
| transcription factor Dp-1 | 3.0 | 0 | 0.0107 |
| aryl hydrocarbon receptor | 3.0 | 0 | 0.0346 |
| glycoprotein (transmembrane) nmb | 3.0 | 0 | 0.0295 |
| chromosome 13 open reading frame 15 | 3.0 | 0 | 0.0319 |
| collagen, type I, alpha 2 | 3.0 | 0 | 0.0293 |
| O-linked N-acetylglucosamine (GlcNAc) transferase | 3.0 | $1.0 \times 10^{-4}$ | 0.0419 |
| B-cell translocation gene 1, anti-proliferative | 3.0 | $1.0 \times 10^{-4}$ | 0.0424 |
| phosphatidic acid phosphatase type 2B | 3.0 | 0 | 0.0339 |
| cyclin-dependent kinase inhibitor 2B (p15, inhibits CDK4) | 3.0 | 0 | 0.0342 |
| TAF10 RNA polymerase II, TATA box binding protein | 3.0 | $1.0 \times 10^{-4}$ | 0.0495 |
| serpin peptidase inhibitor, clade E (nexin, | 2.9 | $1.0 \times 10^{-4}$ | 0.0453 |
| COMM domain containing 2 | 2.9 | $1.0 \times 10^{-4}$ | 0.0363 |
| transducer of ERBB2, 1 | 2.9 | $1.0 \times 10^{-4}$ | 0.043 |

**Table 2.** The top one hundred probesets reaching differential expression of 2.0 or greater and corrected *p*-value less than 0.05 in human macular outer CEC compared with macular inner CEC.

| Gene Title | Fold Change | Uncorrected *p*-Value | Corrected *p*-Value |
|---|---|---|---|
| histone cluster 1, H3b | 14.7 | 0 | 0 |
| LSM4 homolog, U6 small nuclear RNA associated (S. cerevisiae) | 10.2 | 0 | 0 |
| translocase of inner mitochondrial membrane 44 homolog (yeast) | 8.9 | 0 | 0 |
| chromosome 6 open reading frame 108 | 8.4 | 0 | 0 |
| fascin homolog 1, actin-bundling protein (Strongylocentrotus purpuratus) | 7.9 | 0 | 0 |
| FXYD domain containing ion transport regulator 6 | 7.8 | 0 | 0 |
| leucine rich repeat containing 15 | 7.6 | 0 | 0 |
| translocase of inner mitochondrial membrane 44 homolog (yeast) | 7.4 | 0 | 0 |
| RAS and EF-hand domain containing | 7.4 | 0 | 0 |
| kinesin light chain 1 | 6.9 | 0 | 0 |
| RAS and EF-hand domain containing | 6.8 | 0 | 0 |
| cyclin K | 6.4 | 0 | 0 |
| polypyrimidine tract binding protein 1 | 6.0 | 0 | 0.001 |
| valyl-tRNA synthetase | 5.9 | 0 | 0.0011 |
| apelin receptor | 5.4 | 0 | 0.0024 |
| fascin homolog 1, actin-bundling protein (Strongylocentrotus purpuratus) | 5.4 | 0 | 0.003 |
| mitogen-activated protein kinase kinase 2 | 5.1 | 0 | 0.0029 |
| cleft lip and palate associated transmembrane protein 1 | 5.1 | 0 | 0.0018 |
| Hypothetical protein LOC339047 | 4.9 | 0 | 0.0006 |
| RNA pseudouridylate synthase domain containing 3 | 4.7 | 0 | 0.0024 |
| histone cluster 1, H2bf | 4.6 | 0 | 0.0025 |
| transforming growth factor beta 1 induced transcript 1 | 4.5 | 0 | 0.0026 |
| mRNA turnover 4 homolog (S. cerevisiae) | 4.5 | 0 | 0.0027 |
| deoxyribonuclease I-like 3 | 4.4 | 0 | 0.0028 |
| RAN binding protein 3 | 4.4 | 0 | 0.0031 |
| histone deacetylase 5 | 4.4 | 0 | 0.0041 |
| cysteine-rich protein 2 | 4.1 | 0 | 0.0023 |
| cell division cycle 34 homolog (S. cerevisiae) | 4.1 | 0 | 0.0103 |
| TAO kinase 1 | 4.1 | 0 | 0.0023 |
| dicarbonyl/L-xylulose reductase | 4.5 | 0 | 0.0037 |
| collagen, type XV, alpha 1 | 3.9 | 0 | 0.009 |
| coronin, actin binding protein, 1B | 3.9 | 0 | 0.0058 |
| mannose receptor, C type 1 /// mannose receptor, C type 1-like 1 | 3.9 | 0 | 0.0076 |
| DEAD (Asp-Glu-Ala-Asp) box polypeptide 54 | 3.9 | 0 | 0.0081 |
| Na+/H+ exchanger domain containing 1 | 3.9 | 0 | 0.0026 |
| sema domain, transmembrane domain (TM) (semaphorin) 6B | 3.9 | 0 | 0.0049 |
| coronin, actin binding protein, 1B | 3.8 | 0 | 0.0076 |
| carboxypeptidase M | 3.8 | 0 | 0.0073 |
| protocadherin 17 | 3.7 | 0 | 0.0086 |
| GINS complex subunit 4 (Sld5 homolog) | 3.7 | 0 | 0.0051 |
| guanine nucleotide binding protein-like 3 (nucleolar)-like | 3.6 | 0 | 0.0058 |
| leucine rich repeat containing 33 | 3.6 | 0 | 0.0087 |

**Table 2.** *Conts.*

| Gene Title | Fold Change | Uncorrected *p*-Value | Corrected *p*-Value |
|---|---|---|---|
| protein kinase C and casein kinase substrate in neurons 2 | 3.6 | 0 | 0.0048 |
| thimet oligopeptidase 1 | 3.6 | 0 | 0.0088 |
| major histocompatibility complex, class II, DR beta 1 | 3.6 | 0 | 0.0086 |
| hypothetical LOC654433 | 3.6 | 0 | 0.0167 |
| zinc finger protein 688 | 3.6 | 0 | 0.0102 |
| histone cluster 1, H1b | 3.6 | 0 | 0.0089 |
| chromosome 1 open reading frame 93 | 3.5 | 0 | 0.0086 |
| SH3KBP1 binding protein 1 | 3.5 | 0 | 0.0088 |
| cleft lip and palate associated transmembrane protein 1 | 3.5 | 0 | 0.0077 |
| Rho GTPase activating protein 29 | 3.5 | 0 | 0.0094 |
| major histocompatibility complex, class II, DR beta 1 | 3.4 | 0 | 0.0057 |
| MAD1 mitotic arrest deficient-like 1 (yeast) | 3.4 | 0 | 0.0083 |
| exosome component 4 | 3.4 | 0 | 0.0104 |
| YKT6 v-SNARE homolog (S. cerevisiae) | 3.4 | 0 | 0.0087 |
| zinc finger CCCH-type containing 7B | 3.3 | 0 | 0.0087 |
| ATPase type 13A2 | 3.2 | 0 | 0.0117 |
| RAB GTPase binding effector protein 2 | 3.3 | 0 | 0.0085 |
| hepatoma-derived growth factor-related protein 2 | 3.3 | 0 | 0.01 |
| phosphoglucomutase 5 | 3.2 | 0 | 0.0153 |
| matrix metallopeptidase 10 (stromelysin 2) | 3.2 | 0 | 0.009 |
| mitogen-activated protein kinase kinase 2 | 3.1 | 0 | 0.009 |
| endoglin | 3.1 | $1.0 \times 10^{-4}$ | 0.0241 |
| R-spondin 3 homolog (Xenopus laevis) | 3.1 | $1.0 \times 10^{-4}$ | 0.0234 |
| F-box and WD repeat domain containing 5 | 3.1 | $1.0 \times 10^{-4}$ | 0.0285 |
| carbohydrate (keratan sulfate Gal-6) sulfotransferase 1 | 3.1 | 0 | 0.009 |
| lipase, endothelial | 3.1 | 0 | 0.0128 |
| Similar to p40 | 3.1 | 0 | 0.0087 |
| spectrin repeat containing, nuclear envelope 2 | 3.1 | $1.0 \times 10^{-4}$ | 0.0237 |
| kinesin light chain 1 | 3.1 | 0 | 0.0103 |
| homer homolog 3 (Drosophila) | 3.1 | 0 | 0.0102 |
| hypothetical protein LOC286434 | 3.1 | $1.0 \times 10^{-4}$ | 0.0208 |
| exosome component 4 | 3.1 | 0 | 0.0146 |
| glucocorticoid receptor DNA binding factor 1 | 3.1 | 0 | 0.0085 |
| WD repeat domain 4 | 3.0 | 0 | 0.0091 |
| cytochrome b5 reductase 3 | 3.0 | 0 | 0.0085 |
| chromosome 21 open reading frame 45 | 3.0 | 0 | 0.0165 |
| cytochrome P450, family 1, subfamily B, polypeptide 1 | 3.0 | 0 | 0.0182 |
| peter pan homolog (Drosophila) | 3.0 | 0 | 0.0101 |
| cytochrome P450, family 1, subfamily B, polypeptide 1 | 3.0 | 0 | 0.0167 |
| cyclin D1 | 3.0 | $1.0 \times 10^{-4}$ | 0.0319 |
| nasal embryonic LHRH factor | 3.0 | 0 | 0.0084 |
| zinc finger protein 160 | 3.0 | 0 | 0.0153 |
| 3-hydroxy-3-methylglutaryl-Coenzyme A synthase 1 (soluble) | 3.0 | 0 | 0.0102 |
| dipeptidyl-peptidase 9 | 3.0 | 0 | 0.01 |
| small optic lobes homolog (Drosophila) | 3.0 | 0 | 0.0179 |
| splicing factor, arginine/serine-rich 8 | 2.9 | $1.0 \times 10^{-4}$ | 0.0211 |
| Wolf-Hirschhorn syndrome candidate 1 | 2.9 | $2..0 \times 10^{-4}$ | 0.0397 |
| DEAD (Asp-Glu-Ala-Asp) box polypeptide 54 | 2.9 | 0 | 0.0135 |
| dedicator of cytokinesis 6 | 2.9 | 0 | 0.015 |
| lysosomal multispanning membrane protein 5 | 2.9 | 0 | 0.0099 |
| FERM domain containing 3 | 2.9 | $1.0 \times 10^{-4}$ | 0.0191 |
| ankyrin repeat domain 1 (cardiac muscle) | 2.9 | $1.0 \times 10^{-4}$ | 0.0231 |
| ring finger protein 125 | 2.9 | 0 | 0.0186 |
| Hypothetical protein LOC203274 | 2.9 | 0 | 0.0175 |
| sorbitol dehydrogenase | 2.9 | 0 | 0.0119 |
| ATPase family, AAA domain containing 3A | 2.9 | 0 | 0.01 |
| sparc/osteonectin, cwcv and kazal-like domains proteoglycan (testican) 1 | 2.9 | 0 | 0.0103 |
| Hypothetical protein LOC100129502 | 2.9 | $1.0 \times 10^{-4}$ | 0.0209 |

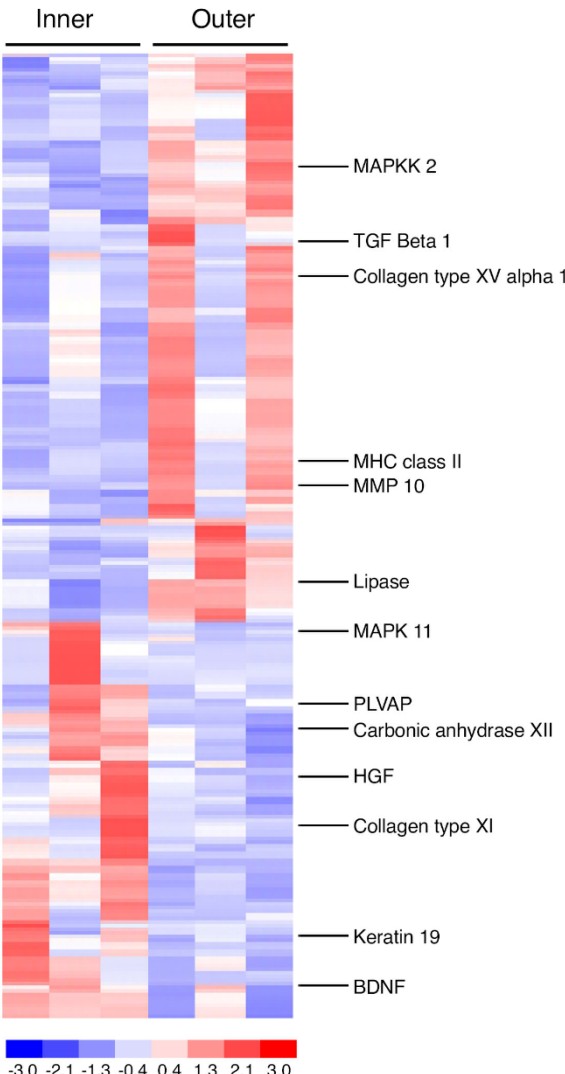

**Figure 2.** Heatmap of the differential probeset expression between human macular inner and outer choroidal microvascular EC reaching statistical significance (adjusted *p* < 0.05). Up-regulated probesets are shown in red, down-regulated in blue. A total of 386 different probesets are represented. The positions of selected probesets thought to be important in endothelial cell biology are shown. The scale bar underneath the heatmap represents the log2 fold change from the mean.

### 3.5. Real-Time PCR

Real-time PCR was used to validate the differences in gene expression between proliferating human inner and outer CEC. Four transcripts were chosen that demonstrated at least a 2-fold differential expression between the EC on microarray analysis and were thought to be relevant to a range of different EC attributes such as the cytoskeleton, intracellular signaling, growth factor and chemokine production. The chosen transcripts were: Keratin 19, BDNF, CXCL 12, and MAPK. Figure 3 demonstrates examples of RT-PCR dissociation curves for Keratin 19 in peripheral inner choroidal, macular inner and macular outer choroidal ECs. The differences in expression between the microarray and real-time PCR techniques were similar for all four transcripts evaluated (Table 3) and demonstrated no significant difference between the results of the two methods for each gene and confirm the overall reliability of the results obtained by the Affymetrix microarray technique.

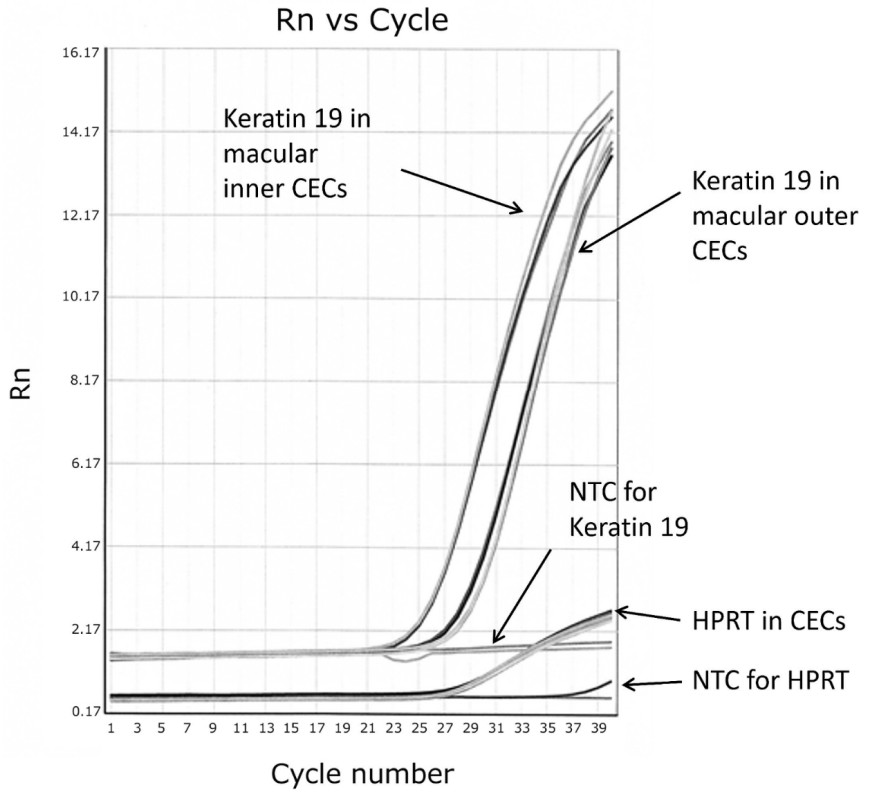

(**A**)

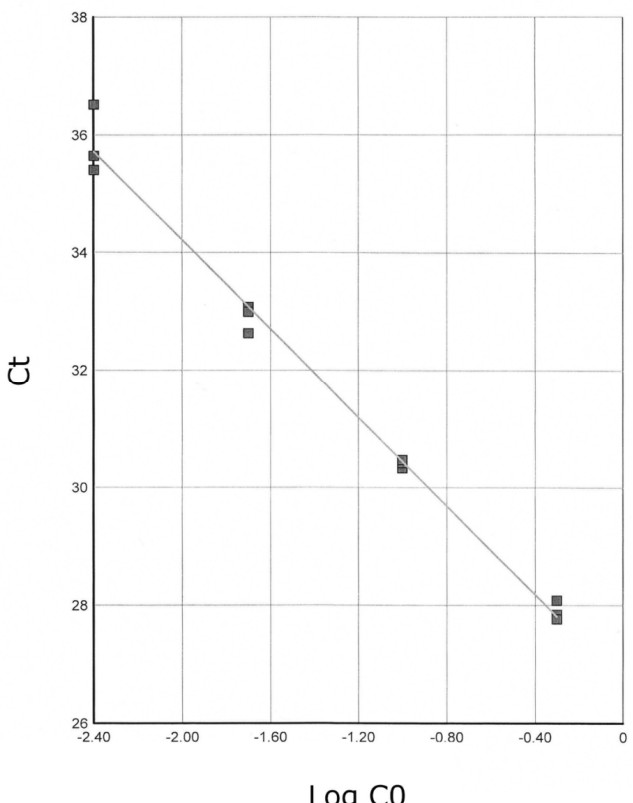

(**B**)

**Figure 3.** *Conts.*

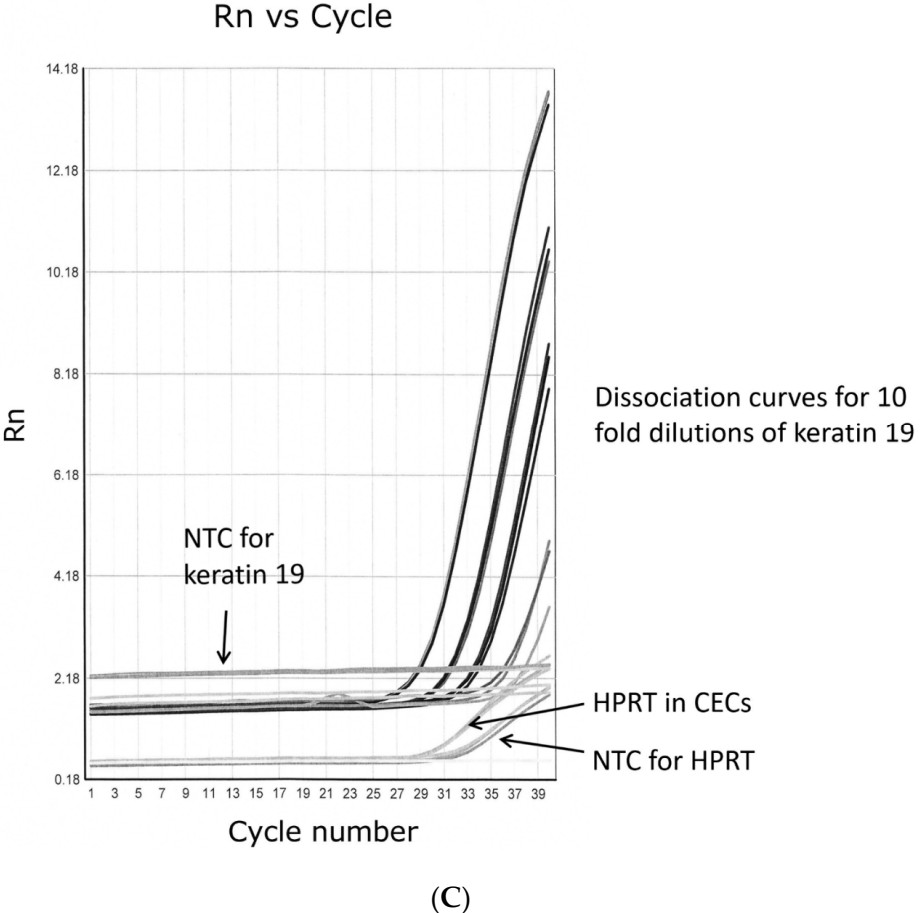

(**C**)

**Figure 3.** (**A**) Examples of the real time PCR dissociation curves for Keratin 19 in the, macular inner and macular outer choroidal ECs. The Y axis depicts the fluorescence of the reporter signal normalised to a reference signal (Rn,) while the X axis depicts the cycle number. The probeset under investigation is represented by all 3 curves of the sample triplicate and also includes a probeset for the reference housekeeping gene, hypoxanthine-guanine phosphoribosyltransferase (HPRT). Additionally represented are the curves of the No Template Control (NTC) (sterile water). (**B**) represents the calculated standard curve derived from Figure 3a demonstrating its linearity of the range tested. (**C**) demonstrates the dissociation curves for a series of 10 fold dilutions for Keratin 19 reference sample.

**Table 3.** Differences in gene expression of selected genes for proliferating human macular inner and outer CEC according to microarray and RT-PCR.

| | | Difference in Gene Expression (Fold Change) | | |
|---|---|---|---|---|
| **Gene Transcript** | **Affy ID** | **Fold Change in Gene Expression Relative to Human Macular Inner CECs** | | |
| | | **Microarray (SD)** | **RT-PCR (SD)** | (***p* Value**) |
| Keratin 19 | 201650_at | 7.3 (1.8) | 10.5 (2.0) | (0.15) |
| Brain derived Neurotrophic factor | 206382_s_at | 8.9 (0.8) | 7.2 (0.6) | (0.34) |
| CXCL 12 | 203666_at | −2.4 (−1.0) | −3.6 (−1.0) | (0.21) |
| MAPK 11 | 206040_s_at | −2.8 (−0.9) | −3.3 (−0.64) | (0.52) |

Ct is the crossing point or threshold at which fluorescence can be detected and log C0 is the log of the relative standard concentration (chosen to correspond to the expected relative concentration of probeset in the samples).

*3.6. Immunohistochemistry*

The expression of *BDNF* and *PLVAP* were compared between inner and outer choroid using immunofluorescent staining with specific antibodies. When visualised using the same microscope imaging settings increased *BDNF* expression could be seen in inner macula choroid compared to outer, as indicated by arrows (Figure 4A). Increased staining was also seen in inner macula choroid with *PLVAP* antibodies (Figure 4E) and this appeared consistently in both macula and peripheral choroid (Figure 4E,F). These results are consistent with the microarray analysis showing that both genes are upregulated in the inner macular CECs.

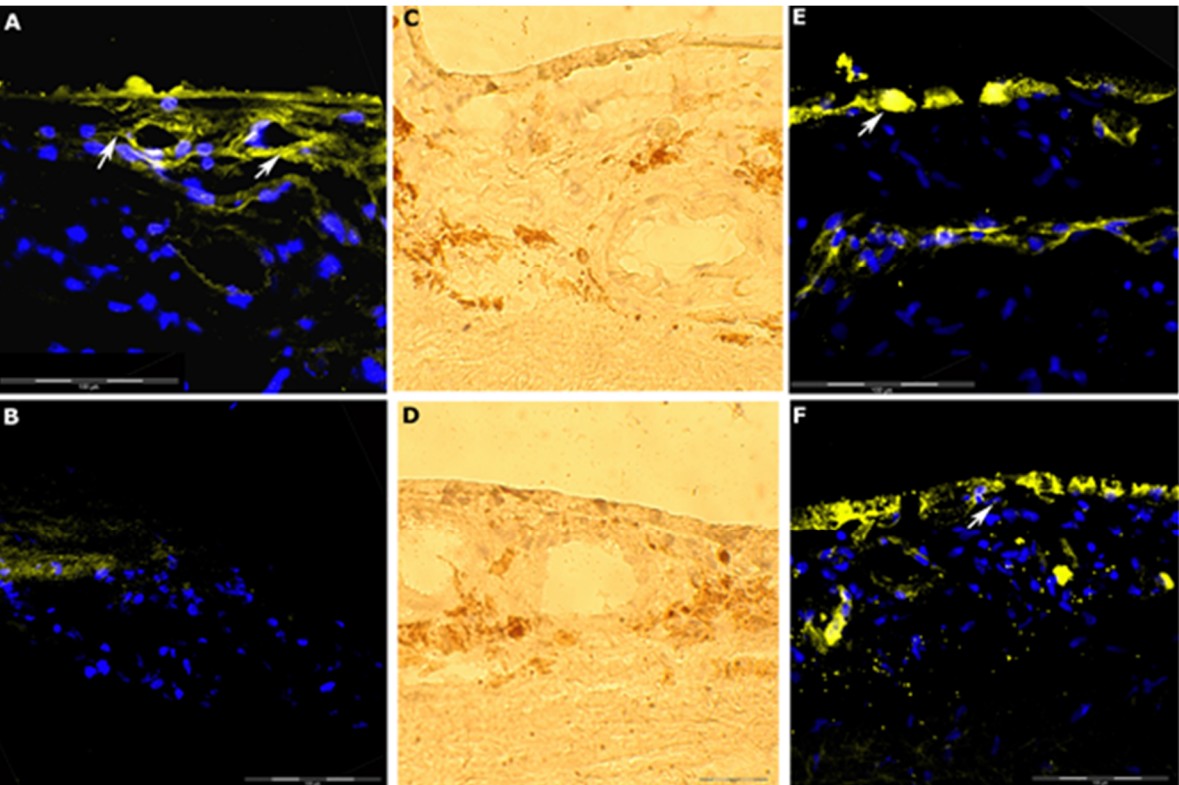

**Figure 4.** Immunofluorescent staining of inner and outer choroid showing increased staining of *BDNF* and *PLVAP*. Macula and peripheral choroid sections were immunofluorescently stained with antibodies to *BDNF* (**A**,**B**) and *PLVAP* (**E**,**F**) (yellow). In inner macula choroid ((**A**) and (**E**), arrows) staining was increased for both BDNF and PLVAP compared to outer choroid, this was also seen in peripheral choroid with *PLVAP* but not *BDNF*. Cell nuclei were counter stained with DAPI (blue). Hematoxylin and Eosin staining of serial sections visualised the tissue structures of macula choroid (**C**) and peripheral choroid (**D**).

## 4. Discussion

It still remains a mystery as to why different ocular posterior segment disorders demonstrate exquisite topographic selectivity for the choroidal vascular bed. Similarly important is the quest for treatments that are selective for the cells involved in the pathological process but which leave normal, juxtaposed cells unaffected—the so-called magic bullet. In the case of wet (neovascular) macular degeneration, this would involve a treatment that specifically targets the proliferating EC of the inner choroid of the macula. While the propensity of the macular for choroidal neovascularisation is unexplained, an animal model of nAMD, using focal laser to apply identical burns to the extra-macular areas, was relatively ineffective at stimulating CNV compared with those placed in the macula, suggesting a predilection of the macula for CNV formation [18].

Over the years, a number of theories have been proposed as to why the human macula is predisposed to choroidal neovacularisation. These include: increased free radical production at the macula caused by the eye's inherent focusing of ultra-violet or visible light at this

location [19–22] although the majority of large scale epidemiological studies have failed to demonstrate an association [23–25], and a localised immune response to preferential drusen deposition at the macula, causing secondary angiogenic events [26]. This latter theory is given weight after Mullins et al. Described differences in the distribution of ICAM 1, a cell surface molecule important in immune response and MHC class I, across the choriocapillaris and retina, with labelling being strongest in the macular area [27,28]. Other hypotheses rely on local differences in Bruch's membrane structure or the thickness of the elastic layer of Bruch's membrane in the macular area which is up to six times thinner and up to five times less abundant than in the periphery [10–12], or local structural differences in the choroid.

While there have been reports on the morphological and gene expression differences in human RPE cells to explain the difference in disease site specificity [2–4], little is known regarding differences in gene expression between peripheral and macular CECs. A number of researchers have used a range of techniques such as microarrays, single cell or localised RNA sequencing and proteomic analysis to establish local gene expression of topographical RPE/choroid complexes or of isolated single cells [29–32]. In 2007, Radeke et al. studied the topographic differences using microarrays, between macular and peripheral choroid/RPE complexes [29] and this demonstrated a difference in 76 probesets Of significance were a number of inflammation related genes such as *CXCL14*, *CCL19* and *CCL26* that were significantly up-regulated in the macula. In 2014, Whitmore used RNA sequencing to study the transcriptome of macular, nasal and temporal PE/choroidal complexes and found distinct differences in the expression of a small number of RPE and endothelial genes [30], while Skeie undertook proteomic analysis of macular and peripheral RPE/choroid complexes and found differences in protein expression between the two [31]. The macula demonstrated increased expression of complement and inflammatory proteins while cells in the periphery show increased expression of anti-oxidants. Unfortunately, all of the samples in these three studies were a homogenates of choroid and RPE and any differences observed could not be ascribed solely to the choroid or, more specifically, the choroidal endothelial cell. In 2019, Voigt used single cell RNA sequencing of CECs in surgically excised choroidal neovascular membranes and found increased expression of the regulator of cell cycle gene (*RGCC*), carbonic anhydrase 4 *(CA4)* and *PLVAP* [32].

The current study did not demonstrate any differences in gene expression between matched human macular and peripheral CEC. This confirms that despite the macular and peripheral choriocapillaris having different ultra-structural appearances, the lining endothelium appears to exhibit the same gene expression generally. This would suggest that the propensity of the macula to suffer CNV is not due to topographical differences in endothelial cells, and that any treatment designed to target proliferating inner CEC is just as likely to affect peripheral ECs as it is the macula. The corollary to this is that treatments that are effective for choroidal vascular disease within the macula area must also be effective in similar diseases occurring in the inner choroid, outside the macula, such as peripapillary and extramacular CNV.

By comparison, greater differences were found between matched inner and outer macular CEC, with 1.6% of probesets showing a significant difference. While no major differences in canonical pathways were discovered and many of the probesets remain unclassified, differences in functional pathways such as nervous system development, cell signalling, immune functions and cell morphology were represented. These differences are likely to be related to their different functions within the choroidal vascular unit.

Perhaps one of the most important probesets found to be differentially expressed is that of *PVLAP* which was up-regulated in macular inner CECs. PVLAP is a major structural protein known to be associated with fenestrations. The choriocapillaris, which is where macular inner CECs reside is known to be fenestrated, whilst the larger calibre outer choroidal vasculature (where macular outer CECs are found) is not. Interestingly, not all fenestrated ECs express PLVAP, with those of the liver and the glomerulus showing negative expression [33]. BDNF is a member of a group of proteins called neurotrophins which promote growth, survival and differentiation of neurones in the central and peripheral nervous system. Within the eye,

BDNF is known to be secreted by RPE cells, photoreceptors and Muller cells and has been shown experimentally to prevent ischaemic ganglion cell death, and protect photoreceptors from light induced toxicity [34]. Recently, BDNF has been found to be secreted by vascular ECs and may be responsible for the levels of the growth factor detected in serum. However, its role outside the central and peripheral nervous system remains unknown [35]. In the current study, BDNF was found to be upregulated 9-fold in macular inner CEC compared with matched outer macular EC. Potential roles for BDNF secretion in this location include its involvement in the maintenance and function of neurones within the choroid, which in turn are involved in regulation of choriocapillaris blood flow, or involvement in outer photoreceptor function by the passage of BDNF across Bruch's membrane.

Interestingly, inner and outer macular CECs appear to demonstrate preferential up-regulation of collagen types XI and XV, respectively. Collagen XI is a fibrillar collagen, mutations in which, have been found in Stickler's syndrome. Collagen XV is a non-fibrillar type of collagen which is found in some EC basement membranes and is thought to facilitate binding to surrounding connective tissue [36]. This would suggest that the basement membranes of the inner and outer CECs are different and are perhaps a reflection of their different roles.

The study shows up-regulation of *VEGF* and *HGF* expression by macular inner CEC compared to the outer CEC. This may be important because VEGF is required to maintain the specialised phenotype of EC within the choriocapillaris (fenestrations). Hepatocyte growth factor (HGF) is another potent endothelial mitogen secreted by cells of mesenchymal origin, including vascular ECs and macrophages. Its structure, very similar to that of plasminogen, contains a heparin binding domain and is secreted by cells in an inactive form [37]. It relies on the action of serine proteases for activation, and activates cells via the c-met receptor. In the eye, it is thought to play a role in corneal development and maintenance of normal corneal structure, and in the maintenance of trabecular meshwork structure [37]. High levels of HGF and its receptor have been demonstrated in the posterior segment of the eye [38–40]. Elevated levels are also found in the vitreous of diabetics and it has been shown to be a potent angiogenic growth factor (greater than VEGF) and may therefore play a role in proliferative diabetic eye disease [41,42]. While there is a large body of evidence regarding the effect of HGF in retinal neovascularisation, its role in choroidal homeostasis and neovascularisation is less well understood. In a rat laser model of CNV, HGF was found to be up-regulated early on in the angiogenic process within the choroid [43]. To the best of our knowledge, no studies on HGF in human choroid, either normal or those with CNV, have been conducted.

In contrast, macular outer choroidal EC demonstrated up-regulation of probesets for the growth factors *CXCL12* (stromal cell derived factor 1 *[SDF-1]*) and *TGF-Beta* SDF-1 is a known mitogen for ECs. It is also involved in the attraction of endothelial progenitor cells to areas of neovascularisation and has been found on histological examination of excised CNV membranes. Bhutto et al. [44] demonstrated expression of SDF-1 and its receptor within the choroidal stroma (as well as the RPE) and suggested that they may be involved in the recruitment of leukocytes and other inflammatory cells to the choroidal stroma as well of endothelial progenitor cells during local wound healing responses, i.e., angiogenesis. However, its differential expression by macular outer CEC was previously unknown.

While this study describes a number of novel findings regarding choroidal endothelial cells, it does have a number of limitations, largely as a consequence of the availability of suitable donor tissue and the difficulty in maintaining parallel cultures of endothelial cells derived from different sites from the same donors. It is recognised that age can have an effect on the differential gene expression of certain ocular tissues such as the retina [45]. While using donors of similar age may allow this effect to be reduced as a cause of variability, using donors with a range of ages from 42 to 62 is important because choroidal neovascularisation can occur at almost any age. In our experiments, the effect of age was reduced by only selecting those genes for further investigation that were significantly upregulated in all donors.

## 5. Conclusions

In summary, this study has demonstrated small, subtle but important differences between matched proliferating human macular inner and outer CEC; however, no significant differences in canonical or functional pathways were found between macular and peripheral inner CEC. This suggests that the small topographical differences in proliferating inner CEC is probably not the cause of the site specific selectivity of neovascular AMD and that this phenomenon is more likely to be due to topographical differences in other ocular cell types or to selective exposure of the macula to a disease causing agent. A higher level of variability was noted, however, in the gene expression of matched macular inner and outer CEC. This differential gene expression would suggest subtle differences in the roles and microenvironments of the two cell types commensurate with the different structures and functions of the inner and outer choroid. These observed differences may assist us in understanding some of the underlying mechanisms of choroidal neovascularisation and provide potential routes for selective intervention to treat the disease.

**Supplementary Materials:** The following are available online at https://www.mdpi.com/article/10.3390/ijtm1020007/s1. Figure S1. RNA degradation plot for the 9 individual arrays. Figure S2. Plot showing the 9 arrays plotted along the first and second principle components. The macula outer CECs from all donors are grouped together, while the macular and peripheral inner CECs from each individual donor are positioned close together.

**Author Contributions:** Conceptualization, A.C.B. and W.M.A., experimentation, A.C.B., E.P.H., D.C.S., S.J.C. and E.A.S.; data analysis, A.C.B., E.P.H., D.C.S., S.J.C. and E.A.S.; writing—original draft preparation, A.C.B. and W.M.A.; writing—review and editing, A.C.B. and W.M.A.; funding acquisition, W.M.A. and A.C.B. All authors have read and agreed to the published version of the manuscript.

**Funding:** This research was funded by a research grant from The British Eye Research Foundation (Fight for Sight UK), grant number BERF 0406, and an unrestricted research grant from Novartis Pharma, grant number GGM13-EO14. The funders had no role in the design of the study; in the collection, analyses, or interpretation of data; in the writing of the manuscript, or in the decision to publish the results.

**Institutional Review Board Statement:** This study was conducted according to the guidelines of the Declaration of Helsinki, and had the approval of the local research ethics committee (Nottingham Q1060301).

**Informed Consent Statement:** Not applicable.

**Data Availability Statement:** Complete data is available at http://www.ncbi.nlm.nih.gov/geo/query/acc.cgi?token=gnavimkwbrarxqf&acc=GSE51246 (accessed on 1 January 2021).

**Acknowledgments:** This research was supported by a research grant from Fight for Sight UK, and an unrestricted research grant from Novartis Pharma.

**Conflicts of Interest:** W.M.A.: Commercial relationships: acted as consultant for Abbvie, Alimera, Allergan Inc, Bayer, Apellis, Novartis, Pfizer, Santen and Thrombogenics, and has undertaken research sponsored by Allergan, Bayer, Boerhinger Ingelheim, No-vartis and Pfizer. He has received speaker fees and travel grants from Allergan, Bausch and Lomb, Bayer, Novartis and Pfizer. All other authors declare no conflict of interest.

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
