# Peer review of "Regional Differences in Gene Expression of Proliferating Human Choroidal Endothelial Cells"

_2673-8937, doi:10.3390/ijtm1020007_

Round 1

Reviewer 1 Report

The work of Browning and colleagues investigates on an important issue in ophthalmology: are differences in gene expression in choroidal endothelial cells responsible for ultrastructure differences and disease predisposition?

The work is important and correctly carried out. In my opinion, the work would benefit from a different data presentation and clarification of a few point.

In the Introduction, the authors report about previous gene expression studies in this field. However, it is difficult to understand on which cell types these studies were performed and with which results.

Then, the authors describe the 3 distinct layers of the choroid: outer, middle, and internal. However, it is not clearly defined which layer they have chosen to isolate endothelial cells for this study.

In the Result section:

  1. I would have appreciated to see a representative staining of the isolated endothelial cells with factor VIII and CD31, the author performed as a control. No indication on how this assay was performed is present in the Material and Methods.
  2. In my opinion, figure 1 and 2 are useless. Table 1 and Table 2 could be better positioned as supplementary data, whereas the link to complete data should be placed in the Material and Methods.
  3. The complete heatmap of the nine samples should be shown as figure 1.
  4. RT-PCR analysis should be shown as figure 2, showing standard deviation and statistical analyses. In the text, the authors should better explain why they decided to validate these 4 genes.
  5. Why immunofluorescence (not immunohistochemistry, that generally indicates paraffin-embedded samples) was performed on the two indicated genes? Are the results consistent with what observed in the Affimetrix assay?

A minor point: a few misspelled names (BDNF, not BNDF), and typing errors are present throughout the text.

Author Response

Thanks. I've attached combined comments to reviews

The work of Browning and colleagues investigates on an important issue in ophthalmology: are differences in gene expression in choroidal endothelial cells responsible for ultrastructure differences and disease predisposition?

The work is important and correctly carried out. In my opinion, the work would benefit from a different data presentation and clarification of a few point.

In the Introduction, the authors report about previous gene expression studies in this field. However, it is difficult to understand on which cell types these studies were performed and with which results.

Then, the authors describe the 3 distinct layers of the choroid: outer, middle, and internal. However, it is not clearly defined which layer they have chosen to isolate endothelial cells for this study.

In the Result section:

  1. I would have appreciated to see a representative staining of the isolated endothelial cells with factor VIII and CD31, the author performed as a control. No indication on how this assay was performed is present in the Material and Methods.

Representative staining of the isolated ECs are provided is our previous manuscripts (Browning et al;, 2005; cited as reference 15 in the current manuscript). We avoided reproducing them in order to avoid duplication. For completeness, representative images of CD31 and vWF staining have been inserted into the manuscript (as new Fig 1a and b).

  1. In my opinion, figure 1 and 2 are useless. Table 1 and Table 2 could be better positioned as supplementary data, whereas the link to complete data should be placed in the Material and Methods.

We have omitted the original Figs 1 & 2 from the main text but kept them as supplementary figures. We’re happy to omit them completely if that is the Editor’s preference. We believe Tables 1& 2 fit best in the main text, and have retained them as such. However, if the Editor prefers, we are happy to oblige.

  1. The complete heatmap of the nine samples should be shown as figure 2.

The heatmap is now shown as Fig 1.

  1. RT-PCR analysis should be shown as figure 2, showing standard deviation and statistical analyses. In the text, the authors should better explain why they decided to validate these 4 genes.

Part of the RT-PCR data is shown as the new Fig 2.  Explanations have been provided in the manuscript as to why the selected genes were validated. Unfortunately, as the reviewer appreciates, it is impossible to validate all the genes that showed significant change. We therefore decided to select a few that reflected different EC functions of interest.

  1. Why immunofluorescence (not immunohistochemistry, that generally indicates paraffin-embedded samples) was performed on the two indicated genes? Are the results consistent with what observed in the Affimetrix assay?

We believe that findings are consistent with Affy/Q PCR results as indicated in the manuscript.

A minor point: a few misspelled names (BDNF, not BNDF), and typing errors are present.

These typos have been corrected

Reviewer 2 Report

Browning et al., investigated regional differences in gene expression of proliferating human choroidal endothelial cells, using microarrays, and the gene expression data validated by qPCR and by immunohistochemistry. Authors found no differences in probe set expression between inner CECs compared with peripheral inner CECs. A 1.6% of probe sets difference was found between matched, un passaged proliferating human macular inner CEC and macular outer CEC from the same donors. A marked homogeneity of human macular and peripheral inner CECs was noted. Authors concluded that gene expression differences in inner CECs are not responsible for the site-specific selectivity of choroidal neovascularization. 

General comments

Microarrays are limited by the transcript IDs on the Genechip, and as such novel transcript may be missed. Methods such as sequencing of single cells would be an alternative to using microarrays. Please comment briefly why microarrays were chosen in this study.

Specific comments

1.Authors need to provide a detailed description of the source of material from which the samples/cells were collected from. This can be included in the opening paragraph of the material and methods section.

  1. It is understandable that authors cite the literature for the procedure for isolation of cells, however, it would be helpful if they can include a brief description of how this was performed in the current study.
  2. In the confirmation of cell purity, how were the cell populations sorted and the different populations collected. How many cells were eventually used for RNA extraction per cell subpopulation?
  3. For microarray analysis, how many samples were used per group? Please reference the kit used for RNA extraction. What RIN value was used as a cut off for sample inclusion for downstream analysis?
  4. In the data analysis, how was the gene fold change determined given that the control samples are not clearly described? Authors state in line 90 that they had a small sample size, but the exact sample size is not described in the methods. Define the data that was exported to Ingenuity Pathway analysis and define the parameters for analysis used in IPA.
  5. For the genes validated by qPCR, how were they selected? Define the genes in lines 96-101 and state the source of primers used. Authors state that the DD Ct method was used to determine relative gene expression, however, the samples are not clearly defined, and the control samples is not clearly stated.
  6. I wonder why only one target was validated by immunohistochemistry. Please justify this selection, given that microarrays produce differentially expressed genes in their thousands.
  7. Since Authors have decided to give details of the microarray preparation process, please describe this accurately. For example, what happened to the extracted RNA? Where any spike in controls added to the samples? What was done to produce the biotinylated cRNA? For how many hours was hybridization performed for? How a bout a brief description of the wash, and stain processes and sample acquisition? Lines 136 to 138, authors write about RNA degradation and claim to show this in Fig1, why is this good and important?
  8. Fig 2. You might want to consider using the transcriptome console software offered by Affymetrix to produce better and more visible 3D PCA plots for better sample separation.

Table 3. Please provide the SD and p values for all the genes assayed, relative to the control.

  1. Fig 4 is inconclusive as no specific signal is observed for both gene targets. The H&E staining should be repeated with fresh staining solutions as the current image is unacceptable. In the revision, include the target and sample type next to the corresponding images to guide the eye of the reader making interpretation of the figures easy.

Minor comments

Check referencing materials and methods, line 77.

Line 78, please state the Helsinki Declaration in full.

Line 96, what is TaqMan real-time PCR16?

Author Response

Thanks. I've attached combined comments to all reviewers' comments

General comments

Microarrays are limited by the transcript IDs on the Genechip, and as such novel transcript may be missed. Methods such as sequencing of single cells would be an alternative to using microarrays. Please comment briefly why microarrays were chosen in this study.

Specific comments

  1. Authors need to provide a detailed description of the source of material from which the samples/cells were collected from. This can be included in the opening paragraph of the material and methods section.

Explanations have been provided regarding the samples and cells. This information was provided in the 1st paragraph of the results section in the original, and revised manuscript.

  1. It is understandable that authors cite the literature for the procedure for isolation of cells, however, it would be helpful if they can include a brief description of how this was performed in the current study.

We have provided brief descriptions of cell isolation relevant to this study.

  1. In the confirmation of cell purity, how were the cell populations sorted and the different populations collected. How many cells were eventually used for RNA extraction per cell subpopulation?

Details of cell purity and numbers have been included in the revised manuscript.

  1. For microarray analysis, how many samples were used per group? Please reference the kit used for RNA extraction. What RIN value was used as a cut off for sample inclusion for downstream analysis?

Further details of the microarray samples and methodology have been added.

  1. In the data analysis, how was the gene fold change determined given that the control samples are not clearly described? Authors state in line 90 that they had a small sample size, but the exact sample size is not described in the methods. Define the data that was exported to Ingenuity Pathway analysis and define the parameters for analysis used in IPA.

The sample size was stated in the original manuscript (1st paragraph of Results section), and has been repeated in Data Analysis section for clarity. Details of data export and analysis are provided. There has been clarification of how the DDCt analysis was undertaken.

  1. For the genes validated by qPCR, how were they selected? Define the genes in lines 96-101 and state the source of primers used. Authors state that the DD Ct method was used to determine relative gene expression, however, the samples are not clearly defined, and the control samples is not clearly stated.

Explanations have been provided in the manuscript as to why the selected genes were validated. Unfortunately, as the reviewer appreciates, it is impossible to validate all the genes that showed significant change. We therefore decided to select a few that reflected difference EC functions of interest. We have provided further details on the samples.

  1. I wonder why only one target was validated by immunohistochemistry. Please justify this selection, given that microarrays produce differentially expressed genes in their thousands.

As previously explained, only selected genes/probesets were validated on account of limited resources. BDNF and PLVAP were selected because of their importance in EC functions, and to complement our RT-PCR data.

  1. Since Authors have decided to give details of the microarray preparation process, please describe this accurately. For example, what happened to the extracted RNA? Where any spike in controls added to the samples? What was done to produce the biotinylated cRNA? For how many hours was hybridization performed for? How about a brief description of the wash, and stain processes and sample acquisition? Lines 136 to 138, authors write about RNA degradation and claim to show this in Fig1, why is this good and important?

Further details of the microarray sample preparation have been provided in the revised manuscript.

  1. Fig 2. You might want to consider using the transcriptome console software offered by Affymetrix to produce better and more visible 3D PCA plots for better sample separation.

We have omitted the original Fig 2 from the revised manuscript. It has been retained as Supplementary information only, if the Editor agrees. Otherwise, we’re happy to delete it completely.

Table 3. Please provide the SD and p values for all the genes assayed, relative to the control.

The SD and p values for the comparison of the QPCR target gene / microarray are now included in table 3.

  1. Fig 4 is inconclusive as no specific signal is observed for both gene targets. The H&E staining should be repeated with fresh staining solutions as the current image is unacceptable. In the revision, include the target and sample type next to the corresponding images to guide the eye of the reader making interpretation of the figures easy.

Unfortunately, we’re unable to provide any further images at this time because of Covid-19 redirection of resources. However, we believe the image provided illustrates our qualitative (rather than quantitative) results adequately.

Minor comments

Check referencing materials and methods, line 77.

Line 78, please state the Helsinki Declaration in full.

Line 96, what is TaqMan real-time PCR16?

Thank you. These points/typos have been addressed.

Round 2

Reviewer 1 Report

The authors revised their paper in agreement with my suggestions

Author Response

thank you

Reviewer 2 Report

I have no additional comments

Author Response

thank you